# Effects of Novel Nutraceutical Combination on Lipid Pattern of Subjects with Sub-Optimal Blood Cholesterol Levels

**DOI:** 10.3390/biomedicines13081948

**Published:** 2025-08-09

**Authors:** Nicola Vitulano, Pietro Guida, Vito Abrusci, Edmondo Ceci, Edy Valentina De Nicolò, Stefano Martinotti, Nicola Duni, Federica Troisi, Federico Quadrini, Antonio di Monaco, Massimo Iacoviello, Andrea Passantino, Massimo Grimaldi

**Affiliations:** 1Cardiology Department, Regional General Hospital ‘F. Miulli’, 70021 Acquaviva delle Fonti, Italy; 2Clinic Pathology Unit, Regional General Hospital ‘F. Miulli’, 70021 Acquaviva delle Fonti, Italy; 3Department of Medicine and Surgery, LUM University “Giuseppe Degennaro”, 70010 Casamassima, Italy; 4Department of Medical and Surgical Sciences, University of Foggia, 71122 Foggia, Italy; 5Istituti Clinici Scientifici Maugeri IRCCS, Cardiac Rehabilitation Unit of Bari Institute, 70121 Bari, Italy

**Keywords:** cardiovascular disease, cardiovascular risk factor, dyslipidemia, low-density lipoprotein cholesterol, nutraceutical

## Abstract

**Background/Objectives**: High concentration of plasma low-density lipoprotein cholesterol (LDL-C) is the predominant cause of atherosclerotic cardiovascular disease progression and coronary heart disease. Nutraceutical combination together with a cholesterol-lowering action provides an alternative to pharmacotherapy in patients reporting intolerance to statins and in subjects with low cardiovascular risk. The effects on lipid parameters were evaluated over 6 months for a food supplement containing aqueous extract of *Berberis aristata* and *Olea europea*, fenugreek seed extract, water/ethanol extract of artichoke leaf and phytosterols from sunflower seeds (Ritmon Colesystem^®^). **Methods**: Laboratory data were obtained at baseline from 44 otherwise healthy subjects (33 males, mean 50 ± 11 years) without cardiovascular disease having LDL-C in the range 115 to 190 mg/dL pharmacologically untreated for hypercholesterolemia. Subjects were re-evaluated at 1, 3 and 6 months during which they took one tablet of Ritmon Colesystem^®^ after dinner. **Results**: At baseline, the mean values were 151 ± 21 mg/dL for LDL-C, 223 ± 24 mg/dL for total cholesterol (T-C), 52 ± 14 mg/dL for high-density lipoprotein cholesterol (HDL-C), and 124 ± 58 mg/dL for triglycerides. A significant reduction in LDL-C was observed; 9 mg/dL (95% confidence interval 3–14), 10 (4–17) and 7 (1–14) at 1, 3 and 6 months. A similar significant trend was detected for T-C while triglycerides did not show significant changes and HDL-C had lower values only at 3 months. **Conclusions**: These nutraceuticals in individuals with sub-optimal blood cholesterol levels at intermediate–low cardiovascular risk reduced LDL-C and T-C over 6 months contributing to the improvement of cholesterol control by dietary supplements.

## 1. Introduction

Cardiovascular diseases account for 32% of global deaths [1]. Atherosclerosis, often linked to aging, is primarily caused by hypercholesterolemia and it is the leading cause of premature death, along with ischemic heart disease [2,3]. Low-density lipoprotein cholesterol (LDL-C) increases cardiovascular risk, whereas high-density lipoprotein cholesterol (HDL-C) decreases it [3]. Clinical evidence supports the effectiveness of lowering LDL-C to prevent cardiovascular events. Studies show that LDL-C is both a high-risk marker and a direct cause of cardiovascular disease [4]. Statins are the most commonly used drugs to treat hypercholesterolemia, reducing LDL-C and major cardiovascular events proportional to baseline risk [3,4]. Poor treatment compliance contributes to variability in LDL-C reduction, with some patients discontinuing treatment long-term [3,5]. The most common adverse effects of statins are muscle cramps, myalgia, or fatigue. Among the strategies aimed at delaying or preventing pathological levels of cholesterolemia, the use of food supplements can allow the improvement of the lipid profile in association with a balanced diet and a healthy lifestyle through the reduction in a sedentary lifestyle in favor of physical movement, reduction in stress, the abolition of alcohol consumption and the habit of smoking [6]. A diet rich in natural antioxidants can play a significant role in preventing atherosclerosis. Natural antioxidants, mainly phenolic compounds, present in food of plant origin and in the human diet are responsible for protection of LDL against oxidation by retarding and preventing foam cell formation, and further minimizing the possible damage of vessels caused by oxidized LDL [7]. The mechanism of antioxidant actions involved either by hydrogen atom transfer, transfer of a single electron, sequential proton loss electron transfer and chelation of transition metals. Phenolic compounds can inhibit pro-inflammatory mediators’ activity or gene expression, up- or down-regulate transcriptional elements involved in antioxidant pathways and reduce pro-inflammatory mediators. The oxidative modification of LDL is the basis of deleterious process of atherosclerosis; thus, its reduction could result in reduced vascular inflammation, oxidative stress and the prevention of platelet aggregation.

This study evaluated a food supplement with five safe ingredients on subjects without cardiovascular diseases and LDL-C levels of 115 to 190 mg/dL, who were not receiving pharmacological treatment for hypercholesterolemia. These ingredients have been evaluated by the scientific community for their effects on dyslipidemia:(1)aqueous extract of *Berberis aristata* (*cortex ex ramis*), titrated at 85% in *berberine*, an alkaloid known for the treatment of hypercholesterolemia, known for its action on the increased expression in the membrane of a receptor protein capable of internalizing LDL-C [8];(2)aqueous extract of *Olea europea* titrated in hydroxytyrosol (SelectSIEVE^®^ OptiChol) which has demonstrated, at a daily dosage of 100 mg, a significant improvement in dyslipidemia in subjects with high cholesterol (115–190 mg/dL) after 1 month of treatment, with a reduction in LDL by 24% [9];(3)fenugreek seed extract (*Trigonella foenum-graecum* L.), an ingredient that may improve dyslipidemia, even in type II diabetic patients [10];(4)water/ethanol extract of artichoke leaf (*Cynara scolymus* L.) titrated at 0.5% in chlorogenic acid, capable of inhibiting the HMGCoA-reductase 16 enzyme and which represents an ingredient with high potential for lowering hypercholesterolemia [11];(5)phytosterols from sunflower seeds (*Helianthus annuus* L.) titrated at 95%, of which 40–50% β-sitosterols, known for their ability to reduce hypercholesterolemia demonstrated in several clinical studies and also recommended by the European Society of Atherosclerosis [12].

The primary aim of the study was to evaluate the effects of a novel nutraceutical combination of these ingredients in otherwise healthy subjects with sub-optimal blood cholesterol on LDL-C levels and lipid pattern as secondary objective.

## 2. Materials and Methods

This experimental single-center and single-arm study enrolled subjects aged 18 to 75 years with LDL-C levels between 115 and 190 mg/dL who were not receiving pharmacological treatment for hypercholesterolemia or who refused statin therapy. The exclusion criteria were as follows: treatment, in the last 2 months, with lipid-lowering, hypoglycemic, anorectic, psychotropic drugs, diuretics, beta-blockers, biologics, steroids, immunosuppressants, other food supplements; triglycerides above 400 mg/dL; history of ischemic heart disease; obesity as Body Mass Index above 30 Kg/m^2^; diabetes mellitus; presence of alterations in thyroid, hepatic or renal function; history of cancer, chronic inflammatory intestinal diseases, malabsorption syndromes, psychiatric diseases, liver cirrhosis, pancreatitis, Human Immunodeficiency Virus; abuse of alcohol and/or drugs; pregnant or breastfeeding women; inability to follow all study procedures.

The study was conducted at the ambulatory service of cardiovascular disease prevention of Cardiology Department, Regional General Hospital ‘F. Miulli’, Acquaviva delle Fonti, Italy. Informed consent was obtained from all patients before inclusion in the study that was executed in accordance with the Declaration of Helsinki after the approval of Ethical Committee “Gabriella Serio” of “IRCCS—Istituto Tumori Giovanni Paolo II di Bari” (number 1472 of 18 December 2023).

At the baseline visit, subjects were encouraged to increase their physical activity; they were given behavioral dietary suggestions to correct unhealthy habits and diet advice by the cardiologist instructing them to follow a Mediterranean diet, avoid excessive intake of dairy products and red meat-derived products during the study and maintain overall constant dietary habits. The subjects were instructed to take one tablet of Ritmon Colesystem^®^ after dinner from the day after the enrolment visit to the end of study (6 months). This food supplement contained aqueous extract of *Berberis aristata* and *Olea europea*, fenugreek seed extract, water/ethanol extract of artichoke leaf and phytosterols from sunflower. None of the extracts contained in the supplement are considered Novel Food (Regulation (EU) 2022/169) and all are suitable for food use. At each post-baseline evaluation, subjects were interviewed about eventual adverse events raised between visits; all unused products were retrieved for inventory and product compliance.

Clinical and laboratory data were collected at baseline, as well as at 1, 3, and 6 months. At each visit, anthropometric measurement and vital signs were recorded, and a plasma sample was obtained after a 12 h overnight fast. Venous blood samples were drawn by a nurse from all patients between 8:00 a.m. and 9:00 a.m. Blood in the 8.5 mL tube (BD Vacutainer^®^ SST™ II Advance serum gel; Becton, Dickinson and Company, Franklin Lakes, NJ, USA) was stored in an upright position for 30 min prior to centrifugation to allow adequate clot formation. Tubes were centrifuged at 2000× *g* for 7 min. Specimens were put onto a GLP automated track system and therefore analyzed in the same manner as routine patient samples. Total cholesterol (T-C), HDL-C, LDL-C, triglycerides, creatinine, aspartate transaminase, alanine transaminase, C-reactive protein (CPR), total and direct bilirubin, urea nitrogen were tested using the Abbott Diagnostics Alinity^®^ ci-series (Abbott Laboratories, Chicago, IL, USA), in a single analytical run. D-Dimers were tested using the Werfen ACL Top750 series. All measurements were performed by trained personnel of the Clinic Pathology Unit at Regional General Hospital ‘F. Miulli’. Estimated GFR was calculated according to Modification of Diet in Renal Disease (MDRD) formula [13].

### Statistical Analysis

Data are reported as mean ± standard deviation, median and interquartile range. Normality was assessed by Shapiro–Wilk test. Categorical data were described as frequencies and percentage. For each parameter, mixed linear regression model was used to evaluate the overall effect over time considering repeated measurements as cluster of observation within patient that was modeled as random effect. The intraclass correlation coefficient (ICC) was used to estimate the proportion of variability attributed to the patient. Baseline characteristics of subjects by 10 mg/dL reduction from baseline in LDL-C at 3 or 6 months were compared with Student’s *t*-test or Mann–Whitney test (continuous variables) and chi-squared test (categorical data). The analyses were carried out using STATA software, version 16 (StataCorp, College Station, TX, USA). A *p* value of <0.05 was considered statistically significant.

We used the standard deviation of the LDL-C change, observed in the placebo group of a randomized, double-blinded, placebo-controlled study that enrolled moderately hypercholesterolemic subjects to treatment with a combined nutraceutical [14]. A total of 44 patients were able to detect, with power of 90% and a significance of 0.05, a variation of at least 10 mg/dL under the hypothesis of standard deviation of 20 mg/dL estimated in the placebo group of a similar study [14]. A reduction in cholesterol of 10 mg/dL reduces the cardiovascular risk at ten years of approximately 0.5% (12.2% to 11.7% in a man and 6.9% to 6.5% in a woman) in a non-smoker subject of 55 years without diabetes mellitus and not treated for hypertension, with 130 mm Hg of systolic blood pressure, 50 mg/dL of HDL-C and T-C from 220 to 210 mg/dL [15]. Secondary end-points are the changes at each time-point from baseline in LDL-C, T-C, HDL-C and triglycerides. Primary and secondary end-points were tested comparing the mean change from baseline than zero. A 95% confidence interval (95%CI) of mean change was graphically described.

## 3. Results

A total of 44 subjects (33 males; aged from 24 to 72 years, mean 50 ± 11) were enrolled from 1 February 2024 to 30 June 2024. Table 1 shows baseline characteristics of subjects. At the enrolment visit, the mean value of LDL-C was 151 ± 21 mg/dL, T-C 223 ± 24 mg/dL, HDL-C 52 ± 14 mg/dL, triglycerides 124 ± 58 mg/dL, CPR 0.20 ± 0.18 mg/L (maximum observed 0.80 mg/L). The estimated 10-year and lifetime risks for atherosclerotic cardiovascular disease was 12.6 ± 11.8% (no subject had diabetes mellitus or hypertension or under hypertension medication).

A total of six subjects missed a single visit: one at 3 months and five at 6 months. During the study, no adverse event was recorded. Table 2 displays laboratory values at 1, 3 and 6 months and their changes from baseline. At the analysis for repeated measurements, by using a mixed linear regression model, a significant change in LDL-C, T-C and HDL-C values was observed (respectively, *p* = 0.002, *p* < 0.001 and *p* = 0.012) but not in triglycerides (*p* = 0.994). The ICC, as measure of proportion of variability explained by the subject, was 0.66 for LDL-C, 0.56 for T-C, 0.84 for HDL-C and 0.53 for triglycerides. Figure 1 shows mean change with 95%CI than baseline in LDL-C (panel A), LDL-C/HDL-C ratio (panel B) and T-C (panel C) at 1, 3 and 6 months. A significant reduction was observed at each time-point in LDL-C (9 mg/dL with 95%CI 3–14 at 1-month, 10 mg/dL with 95%CI 4–17 and 7 mg/dL with 95%CI 1–14, respectively, at 3 and 6 months). The LDL-C/HDL-C ratio had a significant reduction at 1 month (0.11 with 95%CI 0.05–0.22) and 6 months (0.18 with 95%CI 0.03–0.33). Compared to baseline, T-C values were significantly reduced at each evaluation (15 mg/dL with 95%CI 8–22 at 1 month, 18 mg/dL with 95%CI 10–27 and 12 mg/dL with 95%CI 3–22, respectively, at 3 and 6 months). Figure 2 shows the relative reduction at 1, 3 and 6 months to baseline for LDL-C (panel A) and T-C (panel B). No significant changes across evaluations were detected for HDL-C and triglycerides (Figure 3, respectively, panel A and B) with the exception of slightly lower values for HDL-C at 3 months than baseline. All remaining parameters did not show significant difference between post-baseline values.

The 22 (50%) subjects with baseline LDL-C higher than 150 mg/dL, in comparison to those with lower values, had significant post-baseline higher values of LDL-C (*p* < 0.001) and T-C (*p* < 0.001) while HDL-C (*p* = 0.250) and triglycerides (*p* = 0.318) were not different. The LDL-C before the supplement was not associated with change during the study period for LDL-C (*p* = 0.706), T-C (*p* = 0.139) and triglycerides (*p* = 0.623) while HDL-C increased significantly in those with LDL-C above 150 mg/dL (*p* = 0.005).

Table 3 reports baseline characteristics of subjects by at least 10 mg/dL reduction at 3 or 6 months in LDL-C than baseline. No association was detected, including lipid profile parameters, except for Aspartate transaminase that were associated with a greater LDL-C reduction.

## 4. Discussion

This study assessed the 6-month impact of dietary supplements containing extracts from *Berberis aristata*, *Olea europea*, fenugreek seeds, artichoke leaves and sunflower seed phytosterols on lipid profiles in subjects without cardiovascular disease and sub-optimal blood cholesterol levels. In 44 subjects with LDL-C in the range of 115 to 190 mg/dL not taking lipid-lowering drugs, we evaluated lipid parameters at 1, 3 and 6 months after starting the food supplement with the combined nutraceuticals. Both LDL-C and T-C plasma levels significantly decreased at each time-point, indicating improved cholesterol control by dietary supplements.

Among cardiovascular diseases, responsible for more than 4 million deaths in Europe each year, the coronary heart disease is the most common single cause of mortality, resulting in 20% of deaths in women and 19% of deaths in men [16]. Myocardial infarction and ischaemic stroke are clinical manifestations of atherosclerotic cardiovascular disease, a condition associated with multiple exposures having low-density lipoprotein as the most extensively studied [17]. An elevated plasma cholesterol is associated with an increased risk to develop atherosclerosis and, consequently, cardiovascular heart disease [18,19]. High plasma LDL-C concentration is considered the most atherogenic and predominant cause of atherosclerotic cardiovascular disease progression [20].

Cholesterol is an essential component of cell membranes transported to peripheral cells largely by the apoB-containing lipoproteins in plasma. The retention and accumulation of cholesterol-rich apoB-containing lipoproteins within the arterial intima at sites of predilection for plaque formation is the initiation of atherosclerotic cardiovascular disease with progressive dose-dependent development of atherosclerotic plaque [21,22]. The LDL particles, approximately 90% of circulating apoB-containing lipoproteins in fasting blood, are estimated in clinical practice from cholesterol concentration LDL-C that measures the total amount of cholesterol contained in LDL particles [17]. In the majority of clinical studies for assessing cardiovascular risk and for evaluating therapeutic benefit in randomized clinical trials, plasma LDL-C is used as estimate of the concentration of circulating LDL and a measure of the cholesterol mass carried by LDL particles [23]. Our study was powered to detect a variation of at least 10 mg/dL in LDL-C after 3 months of treatment. We observed a significant effect in terms of plasma levels LDL-C reduction after 1, 3 and 6 months in parallel to T-C lowering without a relevant effect on HDL-C and triglycerides that showed normal baseline values in most of the enrolled subjects. The LDL-C/HDL-C ratio showed a significant reduction at 1 and 6 months. The relative change in LDL-C was in the range 4–7% for LDL-C and 5–8% for T-C. At an exploratory analysis for predictors of LDL-C reduction greater than 10 mg/dL, only greater values of Aspartate transaminase were significantly associated with LDL-C improvement. The baseline LDL-C was not related to changes for lipid profile, excluding an improvement in HDL-C in those with higher values before the supplement.

It is important to note that, after 3 months, the baseline atherosclerotic cardiovascular risk according to the multivariable algorithms used to assess 10-year probability of specific atherosclerotic cardiovascular disease events (coronary, cerebrovascular and peripheral arterial disease and heart failure) [15] on the basis of gender, age, systolic blood pressure, T-C and smoking status had a significant absolute reduction of 1.4% (approximately 10% relatively to baseline risk).

Plasma LDL-C lowering is initiated to decrease the risk for cardiovascular disease development; it has been estimated that a reduction of 1 mmol/L (39 mg/dL) is associated with 22% in risk reduction for coronary artery disease [24]. Pharmacological treatment is recommended for the treatment of hypercholesteromia, starting with statin treatment, which reduces endogenous C synthesis and increases LDL-receptor activity [25]. However, LDL-C lowering is highly variable with many patients that may experience statin intolerance with serious side effects leading to the choice of another statin or alternative regimen. Muscular adverse effects, the predominant statin-associated symptoms, are subjective myalgias reported in 1% to 5% patients in randomized controlled trials and in 5% to 20% within observational research [26]. In case the high-dose statin treatment is not well tolerated or in case the LDL-C lowering by a statin does not reach the intended goal for LDL lowering, the statin dose can be lowered and treatment is combined with ezetimibe [25]. Recent data from an observational and prospective study that documented the use of lipid-lowering therapies in patients ≥18 years at high or very high cardiovascular risk across primary and secondary care settings in 14 European countries has showed that more than 20% of patients are not treated irrespectively by risk classification and atherosclerotic cardiovascular disease status, and a proportion higher than a quarter of patients did not reach the LDL-C goal [27].

In individuals with borderline lipid profile levels or with drug intolerance, the use of lipid-lowering nutraceuticals may be considered when the cholesterol control goal is not achieved [28]. A nutraceutical treatment is considered as modification of food composition leading to a lower plasma LDL-C concentration. The nutraceutical combination together with a cholesterol-lowering action, associated with an adequate lifestyle, provides an alternative to pharmacotherapy in patients who report intolerance to statins and in subjects with low cardiovascular risk [29]. Nutraceuticals and their synergetic combinations have demonstrated a beneficial effect in the management of dyslipidaemia. Several nutraceuticals have been shown to positively modulate lipid metabolism while having different functions; plant sterols and soluble fibers may reduce the intestinal assimilation of lipids and increase their elimination; *berberine* and soybean proteins improve the cholesterol uptake in the liver; policosanols, monacolins and bergamot inhibit hydroxy-methyl-glutaryl coenzyme A reductase enzyme action determining the cholesterol hepatic synthesis; red yeast rice and *berberine* play an important role on endothelial dysfunction and psyllium; plant sterols and bergamot have positive effects on LDL subclasses [30]. The nutraceutical approach is based on dietary supplements enriched in phytosterols and phytostanols that competitively reduce the uptake of cholesterol into intestinal micelles, transporting fats and sterols through the intestine [31]. This leads to a reduction in cholesterol absorption and, on average, a 10% reduction in plasma LDL-C [32].

Regarding what has also been described in the literature, in healthy subjects without significant cardiac history, the new finding of LDL-C values is moderately high, that is, when they exceed the levels considered optimal but not so much as to require pharmacological treatment such as statins, which could be beneficial as a first approach to a new nutraceutical compound [33]. Ritmon Colesystem is a food supplement in capsules specifically formulated to help regulate the metabolism of cholesterol and triglycerides thanks to the presence of fenugreek and the functionality of the cardiovascular system thanks to the presence of *Berbera Aristata*. The capsule generally is recommended to take one per day, preferably during or immediately after one of the main meals. It should be taken as part of a varied and balanced diet. Its benefits are regulation of lipid metabolism, support for the functionality of the cardiovascular system, reduction in intestinal absorption of cholesterol and the antioxidant and protective properties of olive and artichoke on blood vessels. The ingredients present in the compound are as follows: fenugreek, *berberine*, an alkaloid isolated from the bark, root and rhizomes of plants of the genus *Berberis phytosterols*, which reduce intestinal absorption of cholesterol, and olive and artichoke, which have antioxidant and protective properties on blood vessels. Fenugreek seeds consist of galactomannan, insoluble fiber and protein, fat and alkaloids. The mechanisms responsible for the beneficial effects of fenugreek are diverse. Galactomannan, the soluble dietary fiber, influences glycemic response by delaying gastric emptying, inhibition of digestive enzymes, increasing the gut motility and optimizing microbiome balance. Moreover, galactomannan acts as a lipid-lowering agent through inhibition of pancreatic lipases action and reduction in hepatic lipoprotein production, which can explain fenugreek’s effect on triglycerides and visceral fat as waist circumference [34,35]. Also, 4-OH-Ile could decrease triglycerides and increase HDL-C by insulin-secreting and insulin-sensitizing effects in hepatic cells and peripheral tissues [35]. Additionally, diosgenin has both insulin secretagogue and sensitizing impact by reducing oxidative stress of beta cells. Diosgenin improves glucose uptake in peripheral tissue by targeting white adipose tissue directly, reducing hepatic lipid accumulation and increasing biliary cholesterol excretion [34,35]. Furthermore, trigonelline can affect adipocyte lipid accumulation with hypoglycemic and hypolipidemic properties [35]. Fenugreek supplementation significantly improves fasting plasma glucose, triglycerides, HDL-C, systolic blood pressure and waist circumference [36]. Artichoke and its bioactive components reduce cholesterol levels, particularly LDL-C. Several studies have demonstrated that artichoke extracts influence lipid metabolism by decreasing the production of cholesterol and endogenous triglycerides by acting on their excretion or redistribution in the organism [37]. Various studies have demonstrated its potential as an anti-inflammatory, antimicrobial and neuroprotective agent due to its phytochemical composition [37]. Although based on studies involving heterogeneous populations and variable artichoke formulations, artichoke seems to exhibit cardiovascular therapeutic potential, supported by its vasorelaxant effects, angiotensin-converting enzyme-inhibitory activity and clinical evidence of improved flow-mediated dilation [38]. *Berberine* belongs to the class of protoberberines found in several plants and has shown LDL-C lowering effects by inhibiting proprotein convertase subtilisin/kexin type 9 at both transcriptional and protein levels [39]. Nutraceutical pill containing *berberine* was found to lower LDL-C by 32% after 6 months of follow-up [40]. A 12-week treatment with a nutraceutical formulation containing *berberine*, chitosan and red yeast rice was effective in lowering plasma non-HDL-C and LDL-C compared with placebo [41]. The effectiveness of *Olea europea* in providing antioxidant benefits is linked to the breaking of peroxide chain reactions or by averting the copper sulfate-induced oxidation of LDL. The existence of these mechanisms was proved using metal-independent oxidative systems and stable free radicals [42].

Our single-arm trial has some relevant limitations. Although the study was sufficiently powered to detect a variation of 10 mg/dL in LDL-C, the lack of a control group, as well as the absence of an active treatment with a standard pharmacological approach, makes difficult the evaluation of the net impact of intervention based on lifestyle advice and dietary supplements. Study results should be interpreted as exploratory to generate formal hypotheses to be tested in a future randomized clinical trial including a placebo or active drug. Subjects in this study included mainly those with sub-optimal blood cholesterol levels without comorbidities; generalization of findings to individuals at high cardiovascular risk should be made with caution. The emerging evidence of the improvement in plasma cholesterol control by nutraceutical approaches requires more experimental research to also better evaluate long-term effects.

## 5. Conclusions

With the growing evidence unequivocally demonstrating a causal relationship between LDL-C levels and cardiovascular events, such as myocardial infarction and stroke, LDL-C has become a crucial therapeutic target in the management of cardiovascular disease. The greater the absolute reduction in C-LDL, the greater the benefit in terms of cardiovascular risk reduction; it is therefore important to have more useful compounds in the therapeutic armamentarium currently available for the reduction in cardiovascular risk, without forgetting to include in the initial approach to dyslipidemia the modification of lifestyle combined with moderate physical activity. The results of this study provide further elements available to the clinician in tailoring the best lipid-lowering therapy according to the patient’s characteristics. In young patients with sub-optimal blood cholesterol levels at intermediate–low cardiovascular risk and free from significant carotid atherosclerosis, nutraceuticals containing extracts of *Berberis aristata*, *Olea europea*, fenugreek seed, artichoke leaf and sunflower phytosterols reduce both total cholesterol and LDL-C levels at 6 months. Further randomized studies are needed to confirm the maintenance of this effect over time, the benefit of these nutraceuticals on lipid patterns also in terms of control in atherosclerosis progression and prevention of cardiovascular disease.

## Figures and Tables

**Figure 1 biomedicines-13-01948-f001:**
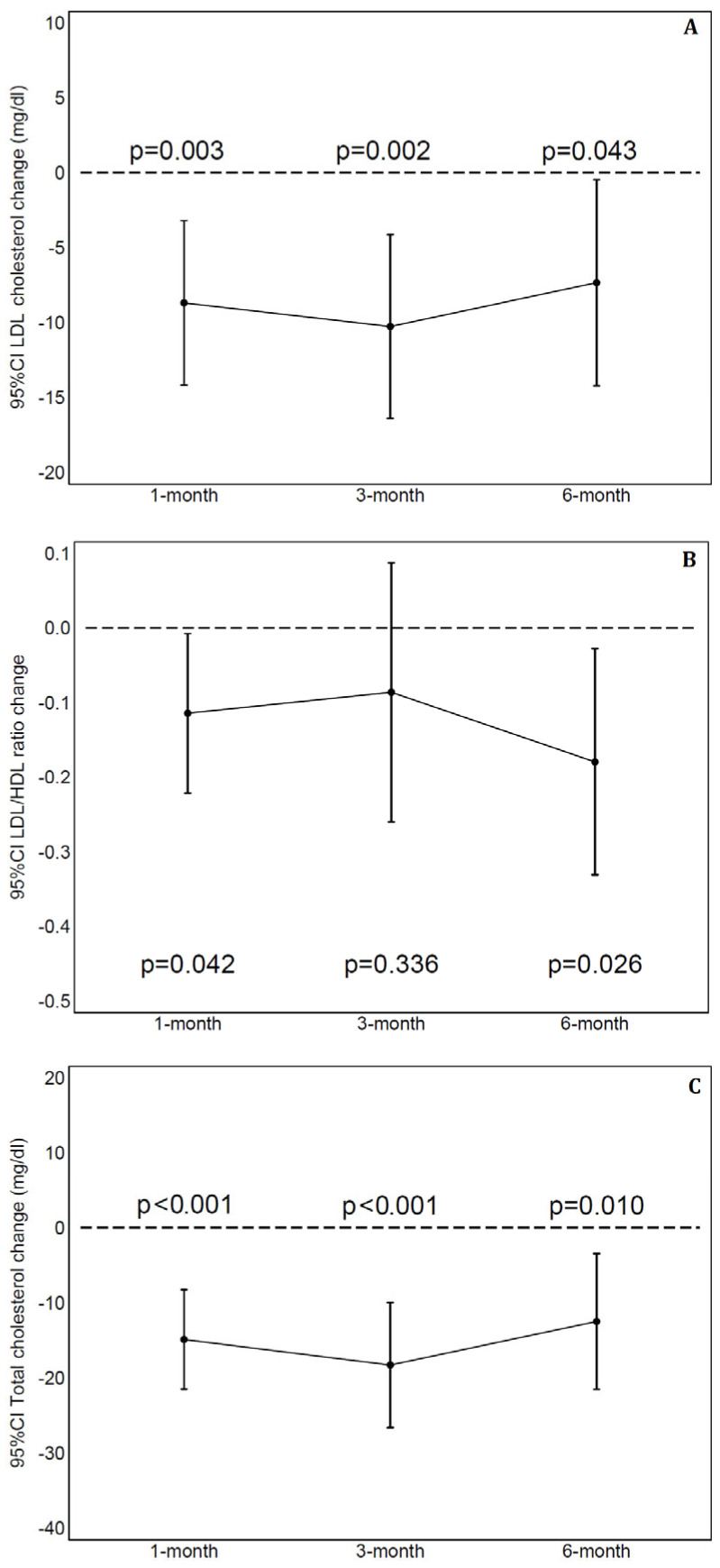
Plasma low-density lipoprotein cholesterol (LDL-C), LDL/HDL ratio and total cholesterol (respectively, panel (**A**–**C**)) change from baseline to 1, 3 and 6 months. 95%CI = 95% Confidence Interval.

**Figure 2 biomedicines-13-01948-f002:**
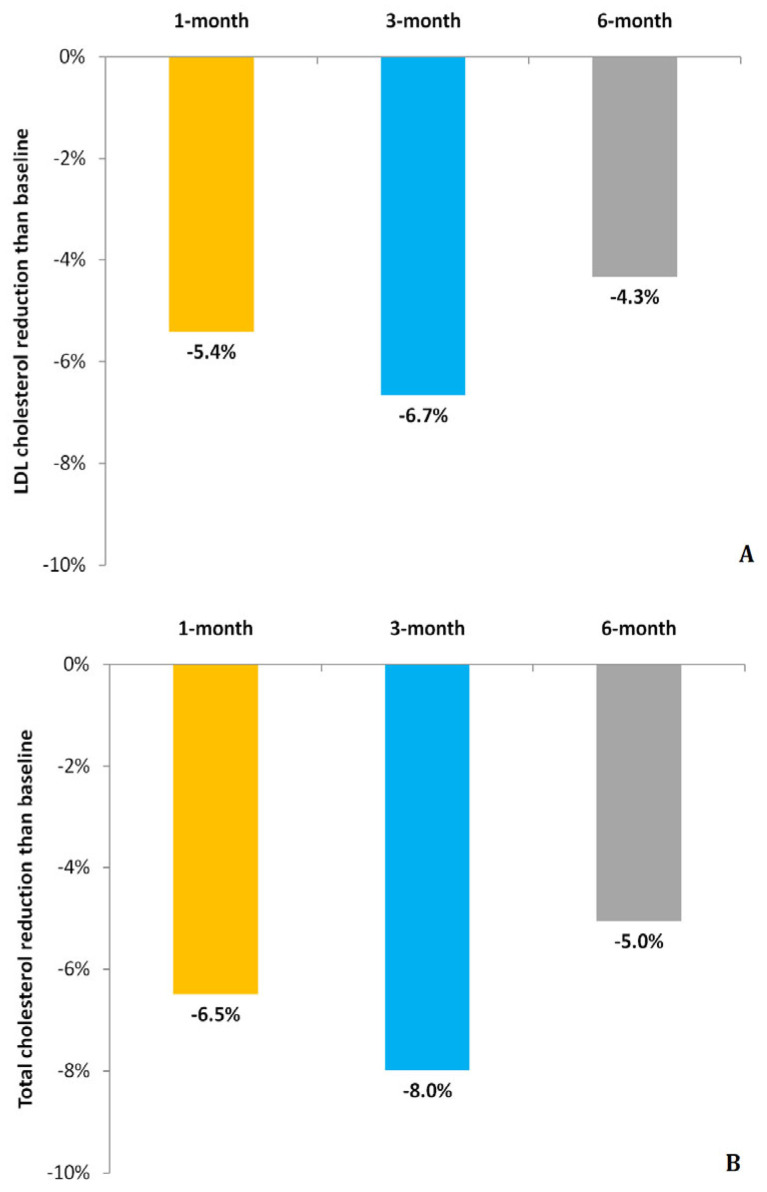
Relative reduction to baseline in plasma low-density lipoprotein cholesterol (LDL-C) and total cholesterol (respectively, panel (**A**) and (**B**)) at 1, 3 and 6 months.

**Figure 3 biomedicines-13-01948-f003:**
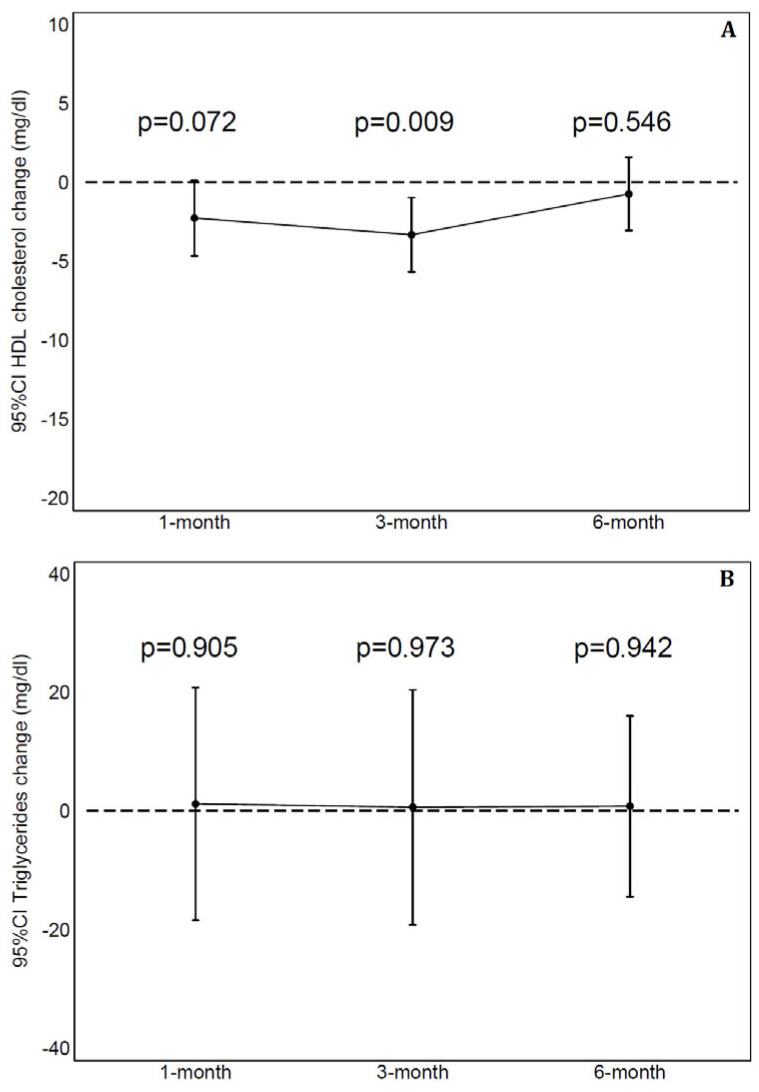
Plasma high-density lipoprotein cholesterol (HDL-C) and triglycerides (respectively, panels (**A**) and (**B**)) change from baseline to 1, 3 and 6 months. 95%CI = 95% Confidence Interval.

**Table 1 biomedicines-13-01948-t001:** Baseline characteristics of 44 subjects enrolled into the study.

	N = 44
Age (years)	54 (43;58)
<50 years	18 (41%)
50–60 years	21 (48%)
>60 years	5 (11%)
Male	33 (75%)
Current smoker	19 (43%)
Weight (Kg)	76 (68;85)
Body Mass Index (Kg/m^2^)	26.0 ± 3.7
Waist circumference (cm)	87 (80;95)
Heart rate (b/min)	68 (60;75)
Systolic blood pressure (mm Hg)	126 ± 16
Diastolic blood pressure (mm Hg)	75 ± 8
Total cholesterol (mg/dL)	223 ± 24
HDL cholesterol (mg/dL)	52 ± 14
LDL cholesterol (mg/dL)	151 ± 21
LDL/HDL ratio	3.2 ± 0.9
Triglycerides (mg/dL)	124 ± 58
D-Dimers (ng/mL)	283 (170;442)
Creatinine (mg/dL)	0.86 ± 0.12
Estimated GFR (mL/min/1.73 m^2^)	100 ± 11
Azotemia (mg/dL)	33 ± 9
Aspartate transaminase (U/L)	22 ± 5
Alanine transaminase (U/L)	24 ± 13
C-Reactive Protein (mg/L)	0.20 ± 0.18
Total bilirubin (mg/dL)	0.81 ± 0.36
Direct bilirubin (mg/dL)	0.240 ± 0.089

Mean ± Standard Deviation or median (Interquartile range). GFR = Glomerular Filtration Rate.

**Table 2 biomedicines-13-01948-t002:** Laboratory values at 1, 3 and 6 months and their changes from baseline in the 44 subjects enrolled into the study.

	1 Month	3 Month	6 Month
Total cholesterol (mg/dL)	208 ± 28	205 ± 32	210 ± 27
Change than baseline (mg/dL)	−15 ± 22	−18 ± 28	−12 ± 29
HDL cholesterol (mg/dL)	49 ± 12	49 ± 11	51 ± 11
Change than baseline (mg/dL)	−2 ± 8	−3 ± 8	−1 ± 7
LDL cholesterol (mg/dL)	142 ± 24	141 ± 27	144 ± 25
Change than baseline (mg/dL)	−9 ± 19	−10 ± 21	−7 ± 22
LDL/HDL ratio	3.0 ± 0.8	3.0 ± 0.9	3.0 ± 0.9
Change than baseline	−0.1 ± 0.4	−0.1 ± 0.6	−0.2 ± 0.5
Triglycerides (mg/dL)	122 ± 72	120 ± 66	114 ± 59
Change than baseline (mg/dL)	1 ± 68	0 ± 68	1 ± 49
D-Dimers (ng/mL)	256 (127;404)	283 (163;478)	307 (162;561)
Change than baseline (ng/mL)	−43 (−110;17)	10 (−78;89)	−22 (−128;163)
Creatinine (mg/dL)	0.87 ± 0.12	0.84 ± 0.12	0.83 ± 0.13
Change than baseline (mg/dL)	0.009 ± 0.073	−0.018 ± 0.081	−0.033 ± 0.085
Estimated GFR (mL/min/1.73 m^2^)	100 ± 10	102 ± 10	102 ± 9
Change than baseline (mL/min/1.73 m^2^)	0 ± 7	2 ± 8	3 ± 8
Azotemia (mg/dL)	35 ± 8	34 ± 9	35 ± 9
Change than baseline (mg/dL)	2 ± 8	1 ± 8	2 ± 8
Aspartate transaminase (U/L)	21 ± 5	23 ± 5	31 ± 52
Change than baseline (U/L)	−1 ± 6	0 ± 5	9 ± 53
Alanine transaminase (U/L)	22 ± 11	20 ± 11	25 ± 21
Change than baseline (U/L)	−2 ± 9	−3 ± 9	2 ± 21
C-reactive protein (mg/L)	0.22 ± 0.22	0.31 ± 0.66	0.22 ± 0.31
Change than baseline in (mg/L)	0.03 ± 0.20	0.13 ± 0.72	0.02 ± 0.24
Total bilirubin (mg/dL)	0.77 ± 0.35	0.76 ± 0.42	0.81 ± 0.53
Change than baseline (mg/dL)	−0.03 ± 0.32	−0.06 ± 0.34	0.01 ± 0.39
Direct bilirubin (mg/dL)	0.25 ± 0.100	0.27 ± 0.11	0.26 ± 0.13
Change than baseline (mg/dL)	0.01 ± 0.10	0.022 ± 0.096	0.02 ± 0.10

Mean ± Standard Deviation or median (Interquartile range). GFR = Glomerular Filtration Rate.

**Table 3 biomedicines-13-01948-t003:** Comparison of subjects’ baseline characteristics by reduction at 3 or 6 months in LDL than baseline.

	<10 mg/dL	≥10 mg/dL	
	n = 22	n = 22	*p*
Age (years)	50 ± 8	50 ± 14	0.606
Males	77%	73%	0.728
Weight (Kg)	77 ± 13	78 ± 17	0.752
Body Mass Index (Kg/m^2^)	26.0 ± 3.1	25.9 ± 4.2	0.976
Waist circumference (cm)	89 ± 11	87 ± 12	0.654
Heart rate (b/min)	69 ± 11	68 ± 6	0.612
Systolic blood pressure (mm Hg)	127 ± 19	124 ± 11	0.759
Diastolic blood pressure (mm Hg)	76 ± 9	75 ± 8	0.574
Total cholesterol (mg/dL)	220 ± 23	225 ± 25	0.557
LDL cholesterol (mg/dL)	147 ± 21	156 ± 20	0.159
HDL cholesterol (mg/dL)	51 ± 14	52 ± 15	0.833
LDL/HDL ratio	3.1 ± 1.0	3.2 ± 0.9	0.700
Triglycerides (mg/dL)	119 ± 48	129 ± 67	0.880
D-Dimers (ng/mL)	259 (147;580)	285 (198;442)	0.766
Creatinine (mg/dL)	0.87 ± 0.13	0.85 ± 0.12	0.620
Estimated GFR (mL/min/1.73 m^2^)	100 ± 10	101 ± 12	0.786
Azotemia (mg/dL)	34 ± 9	32 ± 8	0.429
Aspartate transaminase (U/L)	20 ± 4	25 ± 5	0.002
Alanine transaminase (U/L)	20 ± 5	28 ± 16	0.089
C-reactive protein (mg/L)	0.21 ± 0.21	0.20 ± 0.15	0.187
Total bilirubin (mg/dL)	0.75 ± 0.30	0.87 ± 0.42	0.510
Direct bilirubin (mg/dL)	0.227 ± 0.077	0.255 ± 0.100	0.317

Mean ± Standard Deviation or median (Interquartile range). GFR = Glomerular Filtration Rate.

## Data Availability

The original data presented in this study are included in the article. Further inquiries can be directed to the corresponding author.

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
