# Peer review of "Effects of Novel Nutraceutical Combination on Lipid Pattern of Subjects with Sub-Optimal Blood Cholesterol Levels"

_biomedicines, 2025, doi:10.3390/biomedicines13081948_

Round 1

Reviewer 1 Report

Comments and Suggestions for Authors

This study investigated a nutraceutical formulation in 44 adults with borderline high LDL-C levels over six months. Significant reductions in LDL-C and total cholesterol were observed, while HDL-C and triglyceride levels remained unchanged. The findings suggest potential benefits for individuals intolerant to statins. However, several aspects require clarification.

Considering that LDL-C was reduced by only 7–10 mg/dL, how clinically meaningful is this effect compared to statins or ezetimibe in reducing long-term cardiovascular risk?

Why weren’t baseline LDL-C levels stratified into subgroups (e.g., 115–150 vs. 150–190 mg/dL) to evaluate differential efficacy?

The observation of significantly higher baseline AST in responders raises the question of whether this reflects hepatic metabolism or possible mild hepatotoxicity from the supplement.

Was any statistical correction (e.g., Bonferroni or FDR) applied for multiple comparisons involving lipids and liver enzymes?

Including markers such as ApoB, ApoA1, and the LDL/HDL ratio could have provided a more comprehensive assessment of cardiovascular risk, as they are considered more reliable predictors than LDL-C alone.

Did the authors study the coefficient of variation (CV) for lipid levels to distinguish biological effects from normal intra-individual fluctuations?

Since participants also received lifestyle counseling on diet and physical activity, to what extent can the lipid improvements be attributed solely to the nutraceutical?

Were there any dropouts or missing data ?

How generalizable are these findings to broader populations, particularly those with comorbidities like diabetes, hypertension, or established cardiovascular disease?

Author Response

Reviewer 1

This study investigated a nutraceutical formulation in 44 adults with borderline high LDL-C levels over six months. Significant reductions in LDL-C and total cholesterol were observed, while HDL-C and triglyceride levels remained unchanged. The findings suggest potential benefits for individuals intolerant to statins. However, several aspects require clarification.

R: We thank the reviewer for the comments.

Considering that LDL-C was reduced by only 7–10 mg/dL, how clinically meaningful is this effect compared to statins or ezetimibe in reducing long-term cardiovascular risk?

R: Statins are used to treat hypercholesterolemia, reducing LDL-C and major cardiovascular events. We enrolled subjects with LDL-C levels between 115 and 190 mg/dL. This population, without other risk factors, had a baseline estimated 10-year risks for atherosclerotic cardiovascular disease of 12.6%. Although it is difficult to compare the detected effect to statins or ezetimibe in reducing long-term cardiovascular risk, the estimated risk after 3 months was 1.4% lower than baseline (relative reduction of 10%). This risk reduction may have a value in a population of subjects who were not receiving pharmacological treatment (borderline hypercholesterolemia) or who refused statin therapy.

Why weren’t baseline LDL-C levels stratified into subgroups (e.g., 115–150 vs. 150–190 mg/dL) to evaluate differential efficacy?

R: As showed in Table 3, baseline LDL-C was not different between subjects that reduced LDL-C by 10 mg/dL. We performed a specific analysis as suggested (115-150 vs 151-190 mg/dL). While the absolute values of LDL-C at 1-, 3- and 6-month was higher in those with baseline values of 151-190 mg/dL than 115-150 (p<0.001), no difference was observed in terms of variation than baseline (p=0.706). We reported a comment in the Results and Discussion section.

The observation of significantly higher baseline AST in responders raises the question of whether this reflects hepatic metabolism or possible mild hepatotoxicity from the supplement. Was any statistical correction (e.g., Bonferroni or FDR) applied for multiple comparisons involving lipids and liver enzymes?

R: The baseline values were measured before taking the supplement. Higher baseline AST was associated to wider LDL-C reduction. However, this result has only an exploratory value because not included in primary or secondary end-points. We added a comment in the Discussion section.

Including markers such as ApoB, ApoA1, and the LDL/HDL ratio could have provided a more comprehensive assessment of cardiovascular risk, as they are considered more reliable predictors than LDL-C alone.

R: We thank the reviewer for the suggestion. We were able to analyse the LDL/HDL ratio that showed lower values than baseline. We have included a panel in the Figure 1. Results and discussion section have been modified.

Did the authors study the coefficient of variation (CV) for lipid levels to distinguish biological effects from normal intra-individual fluctuations?

R: We have analysed parameters with statistical tests to distinguish random fluctuations due to sampling technique (no effect of intervention comparing post-baseline values to baseline measurements) than variation correlated to the intervention. Sample size was calculated to detect, with power of 90% and a significance of 0.05, a variation of at least 10 mg/dL. As reported in the result section, most of variability in blood lipid measurements was explained by patient (range 53% to 84%) with a significant contribution of intervention on LDL-C and T-C.

Since participants also received lifestyle counseling on diet and physical activity, to what extent can the lipid improvements be attributed solely to the nutraceutical?

R: As reported in study limitation, the lack of control group make difficult the separation of the two effects.

Were there any dropouts or missing data?

R: All 44 subjects had baseline and 1-month evaluations. At 3-month, 1 subjects did not perform the study visit while 5 other subjects did not perform the last visit. We added this information in the result section.

How generalizable are these findings to broader populations, particularly those with comorbidities like diabetes, hypertension, or established cardiovascular disease?

R: It is difficult to generalize these findings to patients with comorbidities. On the other hand, while pharmacological therapy is an option in individuals at low risk, patients with high cardiovascular risk profile should follow drug intervention strategies as a function of total cardiovascular risk and LDL-C levels. We added a comment in study limitation.

Reviewer 2 Report

Comments and Suggestions for Authors

In this article, the authors described and discussed the effects on lipid parameters over six months of a food supplement containing extract of Berberis aristata and Olea Europea, fenugreek seed, artichoke leaf and phytosterols from sunflower seeds. on 44 healthy patients. As an overall effect, the nutraceutical product significantly reduced the levels of LDL-C and total cholesterol. The topic is important and has a novelty in the point of the continuous growing awareness of the efficacy of the efficacy of nutraceutical as active delivers of substances able to mimic or replace the activity of drugs.

However, there are sever drawbacks that must be clarified:

- A proper use of measure units (mg/dL rather than mg/dl) is advisable 

- “A significant reduction in LDL-C was observed: 8 mg/dl”, please indicate better the reduction respect to the baseline (e.g. a 10% or 5% reduction corresponding to (xxx mg/dL)

- another important issue is the lack of a group treated with standard pharmacological approach

- for the Statistical analysis, please indicate where each test was applied to

- another confounding aspect is the dietary counseling that patients received. The authors should explain why the improvement can be ascribe to the dietary supplement rather than the new lifestyle

- there is also a clear conflict of interest of the authors, as the commercially available nutraceutical product used is Ritmon Colesystem, sold by Dompè, which “partially funded” the study. Why they did not clearly disclose it in the “Conflicts of Interest”? The product should also be indicated with specific trademark, not left without. This issue is quite puzzling.

Author Response

In this article, the authors described and discussed the effects on lipid parameters over six months of a food supplement containing extract of Berberis aristata and Olea Europea, fenugreek seed, artichoke leaf and phytosterols from sunflower seeds. on 44 healthy patients. As an overall effect, the nutraceutical product significantly reduced the levels of LDL-C and total cholesterol. The topic is important and has a novelty in the point of the continuous growing awareness of the efficacy of the efficacy of nutraceutical as active delivers of substances able to mimic or replace the activity of drugs.

However, there are sever drawbacks that must be clarified:

- A proper use of measure units (mg/dL rather than mg/dl) is advisable

- “A significant reduction in LDL-C was observed: 8 mg/dl”, please indicate better the reduction respect to the baseline (e.g. a 10% or 5% reduction corresponding to (xxx mg/dL)

- another important issue is the lack of a group treated with standard pharmacological approach

- for the Statistical analysis, please indicate where each test was applied to

- another confounding aspect is the dietary counseling that patients received. The authors should explain why the improvement can be ascribe to the dietary supplement rather than the new lifestyle

- there is also a clear conflict of interest of the authors, as the commercially available nutraceutical product used is Ritmon Colesystem, sold by Dompè, which “partially funded” the study. Why they did not clearly disclose it in the “Conflicts of Interest”? The product should also be indicated with specific trademark, not left without. This issue is quite puzzling.

R: We thank the reviewer for the suggestions.

We have changed “mg/dl” in “mg/dL” in manuscript and tables as indicated. We have reported the relative reduction respect to the baseline in Results section. In Statistical analysis section , we have indicate the application of each test used. In Study Limitation the absence of control groups and the lack of a group treated with standard pharmacological approach have been commented as well as the possible overlap on lipid changes of dietary counseling that patients received and dietary supplement. We have modified Founding section and added trademark.

Reviewer 3 Report

Comments and Suggestions for Authors

Short summary of the manuscript explaining what the study is about

The authors studied in 44 otherwise healthy subjects the effect of food supplementation by extracts of Berberis aristata, Olea europea, fenugreek seeds,  artichoke leaf and phytosterols from sunflower seeds on on lipid parameters (6 month trial). The authors concluded that the use of these nutraceuticals in individuals with borderline lipid profile levels or with drug intolerance reduced the levels of LDL-C and T-C over 6 months contributing in the improvement of cholesterol control by dietary supplements.

General comments

The manuscript scientifically sound and is the experimental design appropriate. The figures/tables/schemes are appropriate. They properly show the data. The authors used the proper statistical methods. The conclusions consistent with the evidence and arguments presented.

Main issues

In Introduction and Discussion, the protection of LDL-C against oxidation by natural antioxidants (phenolics) in plant extracts must be described.

Biological activity of  plant components must be described in better way.

In Introduction and Discussion, the synergistic effect between individual bioactive compounds from extract must be described.

Minor issues

The title should be written with first capital letters.

Thought entire paper “Berberis aristata” and “Olea europea” must be written with italic.

  1. 1, Abstract, Background/Objectives – It should be “The effect of ….on …was evaluated”.
  2. 1 – The authors’ initials and e-mail addresses.
  3. 2 - It should be “High concentration of low-density lipoprotein cholesterol ….”.
  4. 2 – The plant Latin names must be written with italic.
  5. 2 - It should be “diabetes type 2”.
  6. 2 - It should be “kg/m2”.

Table 4 -It should be “C-reactive protein”

Table 1, 2, 3 – It should be “(kg)”, “Body mas index”, “Estimated GFR”, “C-reactive protein”, “Total bilirubin”, “Direct bilirubin”.

Footnote of Table 2 – Full name of ICC should be cited.

Figure 1 and 2 captions – It should be “low-density lipoprotein cholesterol (LDL-C)  … high-density lipoprotein cholesterol (HDL-C)".

  1. 9, 11, 12 – It should be “Olea europea”.

P.11 - It should be “berberine”, “phytosterols”.

  1. 11 – What does it mean “absorptive enzymes”.
  2. 11 - It should be “pancreatic lipases”.
  3. 11 - It should be “proprotein convertase …”.
  4. 11 - Olea europea do not generate stable resonance …!!!!
  5. 12 – Hydroxyl groups are present not in OLE and HT only in phenolic compounds from OLE and HTP. Explanation of OLE and HT is needed.

Conclusions – Why the authors mentioned “combination of diet and moderate physical activity”?  Physical activity was not included in this investigation.

Comments on the Quality of English Language

Short summary of the manuscript explaining what the study is about

The authors studied in 44 otherwise healthy subjects the effect of food supplementation by extracts of Berberis aristata, Olea europea, fenugreek seeds,  artichoke leaf and phytosterols from sunflower seeds on on lipid parameters (6 month trial). The authors concluded that the use of these nutraceuticals in individuals with borderline lipid profile levels or with drug intolerance reduced the levels of LDL-C and T-C over 6 months contributing in the improvement of cholesterol control by dietary supplements.

General comments

The manuscript scientifically sound and is the experimental design appropriate. The figures/tables/schemes are appropriate. They properly show the data. The authors used the proper statistical methods. The conclusions consistent with the evidence and arguments presented.

Main issues

In Introduction and Discussion, the protection of LDL-C against oxidation by natural antioxidants (phenolics) in plant extracts must be described.

Biological activity of  plant components must be described in better way.

In Introduction and Discussion, the synergistic effect between individual bioactive compounds from extract must be described.

Minor issues

The title should be written with first capital letters.

Thought entire paper “Berberis aristata” and “Olea europea” must be written with italic.

  1. 1, Abstract, Background/Objectives – It should be “The effect of ….on …was evaluated”.
  2. 1 – The authors’ initials and e-mail addresses.
  3. 2 - It should be “High concentration of low-density lipoprotein cholesterol ….”.
  4. 2 – The plant Latin names must be written with italic.
  5. 2 - It should be “diabetes type 2”.
  6. 2 - It should be “kg/m2”.

Table 4 -It should be “C-reactive protein”

Table 1, 2, 3 – It should be “(kg)”, “Body mas index”, “Estimated GFR”, “C-reactive protein”, “Total bilirubin”, “Direct bilirubin”.

Footnote of Table 2 – Full name of ICC should be cited.

Figure 1 and 2 captions – It should be “low-density lipoprotein cholesterol (LDL-C)  … high-density lipoprotein cholesterol (HDL-C)".

  1. 9, 11, 12 – It should be “Olea europea”.

P.11 - It should be “berberine”, “phytosterols”.

  1. 11 – What does it mean “absorptive enzymes”.
  2. 11 - It should be “pancreatic lipases”.
  3. 11 - It should be “proprotein convertase …”.
  4. 11 - Olea europea do not generate stable resonance …!!!!
  5. 12 – Hydroxyl groups are present not in OLE and HT only in phenolic compounds from OLE and HTP. Explanation of OLE and HT is needed.

Conclusions – Why the authors mentioned “combination of diet and moderate physical activity”?  Physical activity was not included in this investigation.

Author Response

The authors studied in 44 otherwise healthy subjects the effect of food supplementation by extracts of Berberis aristata, Olea europea, fenugreek seeds,  artichoke leaf and phytosterols from sunflower seeds on on lipid parameters (6 month trial). The authors concluded that the use of these nutraceuticals in individuals with borderline lipid profile levels or with drug intolerance reduced the levels of LDL-C and T-C over 6 months contributing in the improvement of cholesterol control by dietary supplements.

General comments

The manuscript scientifically sound and is the experimental design appropriate. The figures/tables/schemes are appropriate. They properly show the data. The authors used the proper statistical methods. The conclusions consistent with the evidence and arguments presented.

R: We thank the reviewer for the comments.

Main issues

In Introduction and Discussion, the protection of LDL-C against oxidation by natural antioxidants (phenolics) in plant extracts must be described.

Biological activity of plant components must be described in better way.

In Introduction and Discussion, the synergistic effect between individual bioactive compounds from extract must be described.

  1. We have modified Introduction and Discussion section.

Minor issues

The title should be written with first capital letters.

Thought entire paper “Berberis aristata” and “Olea europea” must be written with italic.

1, Abstract, Background/Objectives – It should be “The effect of ….on …was evaluated”.

1 – The authors’ initials and e-mail addresses.

2 - It should be “High concentration of low-density lipoprotein cholesterol ….”.

2 – The plant Latin names must be written with italic.

2 - It should be “diabetes type 2”.

2 - It should be “kg/m2”.

Table 4 -It should be “C-reactive protein”

Table 1, 2, 3 – It should be “(kg)”, “Body mas index”, “Estimated GFR”, “C-reactive protein”, “Total bilirubin”, “Direct bilirubin”.

Footnote of Table 2 – Full name of ICC should be cited.

Figure 1 and 2 captions – It should be “low-density lipoprotein cholesterol (LDL-C)  … high-density lipoprotein cholesterol (HDL-C)".

9, 11, 12 – It should be “Olea europea”.

P.11 - It should be “berberine”, “phytosterols”.

11 – What does it mean “absorptive enzymes”.

11 - It should be “pancreatic lipases”.

11 - It should be “proprotein convertase …”.

11 - Olea europea do not generate stable resonance …!!!!

12 – Hydroxyl groups are present not in OLE and HT only in phenolic compounds from OLE and HTP. Explanation of OLE and HT is needed.

Conclusions – Why the authors mentioned “combination of diet and moderate physical activity”?  Physical activity was not included in this investigation.

R. We have modified as suggested.

Round 2

Reviewer 1 Report

Comments and Suggestions for Authors

The authors have made substantial changes in several part of the paper and addressed the reviewers’ comments. This manuscript may be accepted for publication.

Reviewer 2 Report

Comments and Suggestions for Authors

The manuscript can now be published.

Reviewer 3 Report

Comments and Suggestions for Authors

The authors corrected this paper properly taken under considerations all my comments. Therefore, I can accept it now.